



# Order of operation for multi-stage post-processing of ensemble wind forecast trajectories

Nina Schuhen[1]

[1]Norwegian Computing Center, P.O. Box 114 Blindern, 0314 Oslo, Norway

**Correspondence:** Nina Schuhen (nina.schuhen@nr.no)

**Abstract.** With numerical weather prediction ensembles unable to produce sufficiently calibrated forecasts, statistical post-processing is needed to correct deterministic and probabilistic biases. Over the past decades, a number of methods addressing this issue have been proposed, with ensemble model output statistics (EMOS) and Bayesian model averaging (BMA) among the most popular. They are able to produce skillful deterministic and probabilistic forecasts for a wide range of applications.

These methods are usually applied to the newest model run as soon as it finished, before the entire forecast trajectory is issued. RAFT (rapid adjustment of forecast trajectories), a recently proposed novel approach, aims to improve these forecasts even further, utilizing the error correlation patterns between lead times. As soon as the first forecasts are verified, we start updating the remainder of the trajectory based on the newly gathered error information. As RAFT works particularly well in conjunction with other post-processing methods like EMOS and techniques designed to reconstruct the multivariate dependency structure

like ensemble copula coupling (ECC), we look to identify the optimal combination of these methods. In our study, we apply multi-stage post-processing to wind speed forecasts from the UK Met Office's convective-scale MOGREPS-UK ensemble and analyze results for short-range forecasts at a number of sites in the UK and the Republic of Ireland.

## 1 Introduction

Numerical weather prediction (NWP) is an inherently uncertain process and even with present-day computational resources, ensembles can not produce perfect forecasts (Buizza, 2018). Statistical post-processing methods have been successfully applied to address these deficiencies, aiming to resolve a multitude of issues. Two important properties of probabilistic forecasts are calibration and sharpness (Gneiting et al., 2007). Calibration is the statistical consistency between the forecasts and the observations, sharpness refers to the amount of predictive uncertainty and thus the extent of information contained in the

forecast. Usually, NWP ensembles lack calibration, as they can not consider all sources of atmospheric uncertainty, but are quite sharp. The main goal of any statistical post-processing process should therefore be to maximize the forecast's sharpness, subject to it being calibrated (Gneiting et al., 2007).



Well-established techniques like ensemble model output statistics (EMOS; e.g., Gneiting et al., 2005) or Bayesian model averaging (BMA; e.g., Raftery et al., 2005) are now available for a number of weather variables, for an overview see Wilks (2018). They measure the ensemble's performance over a training period, either consisting of a rolling window of a few weeks or over a longer, fixed period of time, and then apply a statistical correction to the newest NWP model run. The updated forecasts are usually in the form of a predictive probability distribution, as close to perfect calibration as possible. As EMOS has been proven to work well for our data set, the MOGREPS-UK ensemble produced by the UK Met Office, and is computationally more efficient, we prefer it over BMA.

During the application of some of the methods mentioned above, any physical, spatial and temporal dependency structure from the NWP model is lost and additional effort is needed to restore these patterns (Schefzik and Möller, 2018). In some cases, parametric models can be developed (e.g., Schuhen et al., 2012; Feldmann et al., 2015), however if this is not feasible, techniques like ensemble copula coupling (ECC; Schefzik et al., 2013) and the Schaake shuffle (Clark et al., 2004) provide a non-parametric approach based on reordering samples from the calibrated predictive distributions. In this study, we choose ECC over the Schaake shuffle, as it does not require any additional historical data.

Recently, Schuhen et al. (2019) proposed a new kind of post-processing method, rapid adjustment of forecast trajectories (RAFT), designed to minimize forecast errors on-the-fly. Instead of running once, like EMOS or BMA, between the NWP model run finishing and the publication or delivery of the forecasts, it is applied repeatedly at every lead time step. RAFT works in concert with conventional post-processing techniques and utilizes the error information from the part of a forecast trajectory where observations are already available in order to improve the mean forecast skill for the rest of the trajectory. This means that e.g., any systematic forecast error in a model run that was not picked up by the standard post-processing can now be corrected quickly, once it is recorded. In this way, older forecasts become more valuable and typically outperform the first few forecasts of a new model run. While Schuhen et al. (2019) adjust the deterministic mean forecast only, we will show in this paper how RAFT can also be used to adjust the predictive variance. In general, RAFT applies to any kind of forecast scenario, from the short range to seasonal forecasting, as long as there is sufficient correlation between the errors at different lead times.

With an abundance of post-processing methods available, the question arises in which order they should be employed. Li et al. (2019) look at this problem in the context of generator-based post-processing (GPP; Chen and Brissette, 2014), producing discrete, auto-correlated time series, and dependence reconstruction methods like ECC. When working with EMOS, it should generally be run first in order to remove large-scale calibration errors and provide a skillful baseline forecast. However, it is not obvious how to combine ECC and RAFT. Therefore it is our aim to find the optimal order of operation for these three post-processing methods, each designed to achieve a different objective. The combinations of post-processing methods will be applied to site-specific instantaneous wind speed forecasts produced by the high-resolution MOGREPS-UK ensemble and will be assessed using multiple univariate and multivariate verification tools.

The remainder of this paper is organized as follows. In Section 2, we introduce the data set used in this study. Section 3 describes the individual post-processing methods, including the new RAFT approach, and Section 4 outlines the set of verification metrics which we apply to determine the forecast performance. In Section 5, we illustrate how the different techniques work





by means of an example forecast and present results for the selected combinations of post-processing methods. We conclude with a discussion in Section 6.

## 2 Data

The 10m instantaneous wind speed forecast data used in this study was produced by the UK Met Office's limited-area ensemble MOGREPS-UK (Hagelin et al., 2017). MOGREPS-UK is based on the convection-permitting NWP model UKV, but with a lower resolution of 2.2km. Until March 2016, the global ensemble MOGREPS-G produced both initial and boundary conditions for MOGREPS-UK, but since then initialization has been provided by the UKV analysis.

We use data from all model versions between January 2014 and June 2016, during which the ensemble was initialized four times a day and consisted of 12 members, one control and 11 perturbed forecasts. The perturbations are taken from the global ensemble and combined with analysis increments. Here, we only look at the model run started at 15 UTC, as it was observed in Schuhen et al. (2019) that all four runs behave somewhat similar in terms of predictability. Forecasts are produced for every hour up to 36 hours, covering the short range.

For both estimation and evaluation, SYNOP observations from 152 sites in the British Isles are used (see Fig. 1). To match the observation locations, the forecasts were interpolated from the model grid and subjected to Met Office post-processing in order to correct for local effects and differences between the model and the location's orography. We separate our data set into two parts: the first 12 months are used for estimating the RAFT coefficients and the remaining 18 months for evaluating the post-processing techniques.

## 3 Post-processing methods

In this paper, several post-processing methods are used in various combinations. They all fulfill different purposes: EMOS (ensemble model output statistics) functions as a baseline for producing calibrated and sharp probabilistic forecasts, ECC (ensemble copula coupling) transfers the physical dependency structure of the ensemble to the EMOS forecasts and RAFT (rapid adjustment of forecast trajectories) continually improves the EMOS deterministic forecasts after they have been issued, based on previously not available information.

### 3.1 Ensemble model output statistics

In a first step, all forecasts are post-processed with EMOS, sometimes also called NGR (non-homogeneous Gaussian regression), in order to correct deterministic and probabilistic biases the raw ensemble might suffer from. These deficiencies are a result of the limits of ensemble forecasting in general, as e.g., the ensemble members can only represent a small subset of the multitude of all possible or probable states of the atmosphere at any given point in time. Thorarinsdottir and Gneiting (2010) propose an application of the EMOS method for wind speed forecasts based on truncated Gaussian distributions, although they study maximum instead of instantaneous wind speed.

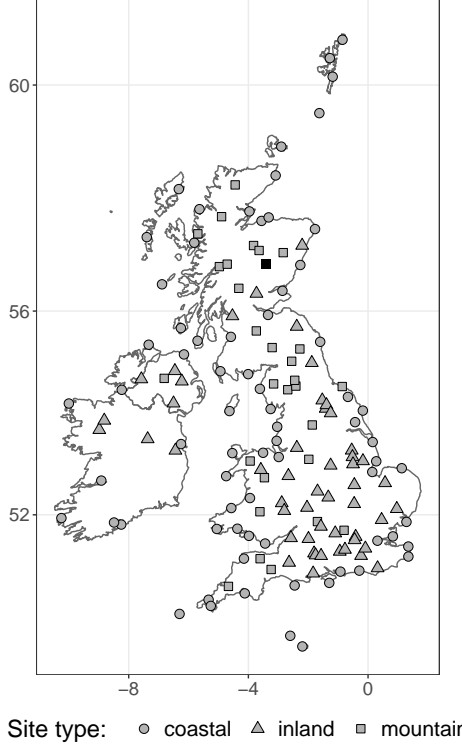

**Figure 1.** Map of the British Isles with the 152 observation locations used in this study. The sites are divided into three categories, coastal, inland and mountain sites, depending on their location and altitude. The black square marks The Cairnwell, a mountainous site in the Scottish Highlands.

As we will see in Sect. 5, this approach (here called gEMOS) produces nearly calibrated forecasts, but they are still slightly underdispersive. For this reason, we investigate a second variant of EMOS introduced by Scheuerer and Möller (2015), logE-
90 MOS, where the predictive distributions are truncated logistic. Due to its heavier tails, the logistic distribution can provide a better fit to the instantaneous wind speed data at hand. Further case studies including various versions of EMOS have shown that sharp and calibrated forecasts can be produced for a number of different NWP ensembles (e.g., Feldmann et al., 2015; Scheuerer and Büermann, 2014; Kann et al., 2009).

Let $X_1, \ldots, X_{12}$ denote an ensemble forecast valid at a specific time and location and $Y$ be the corresponding observed wind
speed. Then we model the gEMOS forecast as a truncated Gaussian distribution with cut-off at zero, in order to account for the non-negativity of the wind speed values:

$$Y \mid X_1, \ldots, X_{12} \sim \mathcal{N}^+ \left( \mu, \sigma^2 \right) \tag{1}$$

Due to the truncation, the negative part of the distribution is cut off and a corresponding probability mass added to the positive part. This means that the parameter $\mu$ here is not the mean of the distribution, but the location parameter. Using the ensemble





mean $\bar{X} = \frac{1}{12}\sum_{i=1}^{12} X_i$ and variance $S^2 = \frac{1}{12}\sum_{i=1}^{12}\left(X_i - \bar{X}\right)^2$ as predictors for the EMOS location parameter $\mu$ and variance $\sigma^2$, we define the following equations for the moments of the predictive distribution:

$$\mu = a + b^2 \cdot \bar{X} \tag{2}$$

$$\sigma^2 = c^2 + d^2 \cdot S^2 \tag{3}$$

The coefficients $b$, $c$ and $d$ are squared in order to simplify interpretability and to make sure that the variance is positive. Minimum score estimation is a versatile way to obtain parameter estimates in such a setting (Dawid et al., 2016). The proper score we want to optimize is the continuous ranked probability score (CRPS; Matheson and Winkler, 1976; Gneiting and Raftery, 2007), which addresses both important forecast properties, sharpness and calibration (for details see Sect. 4). We process all locations and lead times separately, equivalent to the local EMOS approach in Thorarinsdottir and Gneiting (2010), and the training data consists of a rolling period of 40 days. In practice, this means that the training period contains forecast-observation pairs from the last 40 days preceding the start of the model run, valid at the same lead time and location.

In the case of logEMOS, we substitute the truncated Gaussian distribution in Eq. (1) with a truncated logistic distribution:

$$Y \mid X_1, \ldots, X_{12} \sim \mathcal{L}^+\left(\mu, s\right), \tag{4}$$

where $\mu$ is again the location parameter and $s = \sqrt{3\sigma^2}\cdot\pi^{-1}$ the scale. The location parameter and variance are linked to the ensemble statistics in the same way as with gEMOS. Scheuerer and Möller (2015) provide a closed form of the CRPS for a truncated logistic distribution, meaning that gEMOS and logEMOS are comparable in terms of computational cost and complexity. We found parameter estimation to be more stable when applying EMOS to wind speed in knots as compared to meters per second. The ensemble members are treated as exchangeable, in that we use the ensemble mean as predictor for the EMOS location parameter. This results in more robust parameters and faster computation.

### 3.2 Ensemble copula coupling

While EMOS is particularly adept at calibrating ensemble forecast, the ensemble's rank structure is lost in the process. To restore the physical dependencies between forecasts at different lead times, we employ ensemble copula coupling (ECC; e.g., Schefzik et al., 2013). This method makes use of the original ensemble's multivariate dependency information and transfers it to the new, calibrated forecasts.

First, we draw samples from the univariate EMOS distributions. There are several options, but Schefzik et al. (2013) (as a consequence of the discussion in Bröcker, 2012) recommend using equidistant quantiles, as they best preserve the calibration of the univariate forecasts. Then we reorder the quantiles according to the order statistic of the ensemble members. Thus, for each ensemble member $X_i$ at any given forecast lead time $l = 1, \ldots, 36$, we note its rank among the other ensemble members $X_1^{(l)}, \ldots, X_{12}^{(l)}$. We obtain a permutation $\tau_l$ of the numbers $1, \ldots, 12$ such that

$$X_{\tau_l(1)}^{(l)} \leq X_{\tau_l(2)}^{(l)} \leq \cdots \leq X_{\tau_l(12)}^{(l)}. \tag{5}$$





Any ties are resolved at random. Then we apply $\tau_l$ to the EMOS quantiles $\widetilde{X}_1^{(l)}, \ldots, \widetilde{X}_{12}^{(l)}$ and reorder the individual ensemble members so that we obtain a multivariate ensemble

$$\left[\widetilde{X}_{\tau_l(1)}^{(1)}, \ldots, \widetilde{X}_{\tau_l(1)}^{(36)}\right], \ldots, \left[\widetilde{X}_{\tau_l(12)}^{(1)}, \ldots, \widetilde{X}_{\tau_l(12)}^{(36)}\right]. \tag{6}$$

The new ensemble has the same univariate properties as the original EMOS quantiles, as only the order of the ensemble members has changed. However, when we evaluate it using multivariate scores and verification tools, we can see the benefit of

135 ECC. It is a computationally efficient and straightforward method to preserve spatial and temporal features of the NWP model. ECC has been used in a variety of atmospheric and hydrological forecasting scenarios, e.g., Schuhen et al. (2012), Hemri et al. (2015) and Ben Bouallègue et al. (2016).

### 3.3 Rapid adjustment of forecast trajectories

RAFT is a new technique that can be used in conjunction with established approaches like EMOS and ECC. However, it

operates on a different time scale. While EMOS and ECC are applied once when the NWP model run has finished, RAFT continually updates the forecast after it has been issued, using information from the part of the forecast trajectory that has already verified. Essentially, RAFT applies to any weather variable, therefore we do not have to make many alterations to the method for temperature described in Schuhen et al. (2019). We treat all locations separately, as the local error characteristics vary greatly.

In this paper, there are two different RAFT concepts used: we call the standard method that adjusts the EMOS mean RAFT$_\text{m}$, while RAFT$_\text{ens}$ applies to individual ensemble members drawn from the EMOS distribution. RAFT$_\text{m}$ therefore can only improve the deterministic forecast skill, whereas RAFT$_\text{ens}$ provides an adjusted empirical distribution spanned by the ensemble. Both RAFT variants are based on the correlation between observed forecast errors at different lead times. We define the error $e_{t,l}$ at a particular lead time $l$, generated from a model run started at time $t$, as the difference between the forecast and the

observation $y_{t+l}$:

$$e_{t,l} = y_{t+l} - m_{t,l} \tag{7}$$

$$e_{t,l}^{(i)} = y_{t+l} - x_{t,l}^{(i)}, \quad i = 1, \ldots, 12 \tag{8}$$

Equation (7) refers to the RAFT$_\text{m}$ approach, where $m_{t,l}$ is the mean of the EMOS distribution. For RAFT$_\text{ens}$, we need to calculate the error for every ensemble member $x_{t,l}^{(i)}$ (Eq. (8)). To obtain the mean of the truncated Gaussian distribution from

155 the location parameter $\mu$ and spread $\sigma$, we use the following relationship:

$$m = \mu + \sigma \cdot \varphi\left(-\frac{\mu}{\sigma}\right) \cdot \left(1 - \Phi\left(-\frac{\mu}{\sigma}\right)\right)^{-1} \tag{9}$$

The functions $\varphi$ and $\Phi$ here denote the density and cumulative distribution function of the standard Gaussian distribution, respectively. Similarly, the mean of the truncated logistic distribution is

$$m = s \cdot \log\left(1 + \exp\left(\frac{\mu}{s}\right)\right) \cdot \left(1 - \Lambda\left(-\frac{\mu}{s}\right)\right)^{-1}, \tag{10}$$





where $\mu$ is the location parameter, $s$ the scale and $\Lambda$ the CDF of the standard logistic distribution.

From the forecast errors $e_{t,l}$ and $e_{t,l}^{(i)}$, we generate the Pearson correlation coefficient matrix to establish the relationship between the 36 lead times. The left column in Fig. 2 shows the gEMOS error correlation matrix for the weather station on The Cairnwell mountain in the Scottish Highlands. The top plot refers to $\text{RAFT}_{\text{m}}$, while the bottom pictures the correlation for one member of the $\text{RAFT}_{\text{ens}}$ ensemble. All correlations shown are statistically significant at the 90% level. There is a good correlation between a sizeable number of lead times, which makes it possible to define an adjustment period for each lead time, telling us at what point in time to begin with the RAFT adjustments. While the adjustment period applies, we know that a previously observed error $e_{t,l^*}$ at lead time $l^* < l$ gives us reliable information about the future error $e_{t,l}$.

The RAFT model used to obtain the estimated future error $\hat{e}_{t,l}$ at $l = 1, \ldots, 36$ is based on linear regression with the observed error $e_{t,l^*}$ as predictor:

$$\hat{e}_{t,l} = \hat{\alpha} + \hat{\beta} \cdot e_{t,l^*} + \varepsilon, \tag{11}$$

where the random error term $\varepsilon$ is normally distributed with mean zero. The regression coefficients $\hat{\alpha}$ and $\hat{\beta}$ are determined using least squares. Once we have estimated the RAFT regression coefficients for every possible combination of lead times $l$ and $l^*$, we can establish the length $p$ of the individual adjustment periods by looking at those combinations where the estimate of the coefficient $\hat{\beta}$ is significantly greater than zero, meaning that $e_{t,l^*}$ is likely to provide useful information for the prediction of $e_{t,l}$. In order to account for potential limitations in real-time availability of observations, the RAFT adjustments performed at a certain lead time $l - 1$ for any lead time greater or equal to $l$ use the observation recorded at $l - 2$. For the first few lead times of a model run, where no previous error can be computed, the predictors in Eq. (11) are based on the forecasts from the model run initialized 24 hours earlier.

The algorithm for determining the adjustment period corresponds to the one described in Schuhen et al. (2019). In general, it can be applied to any weather variable with errors on a continuous scale. However, it is somewhat arbitrary and can certainly be optimized for individual forecasting scenarios. We proceed as follows:

1. Estimate the regression coefficients in Eq. (11) for all predictors $e_{t,l^*}$ with $l^*$ in $[l - 23; l - 2]$. If any $l^*$ are negative, we use $l^* + 24$ instead.

2. (a) Find the earliest $l^*$ in $[l - 11; l - 2]$, such that the coefficient $\hat{\beta}$ is significantly different from zero at the 90% level for each lead time $l^* + 1, \ldots, l - 2$. Denote the result as $l_p$.

    (b) If there is no result in the previous step, find the earliest $l^*$ in $[l - 19; l - 12]$, such that $\hat{\beta}$ is significantly different from zero at the 95% level for each lead time $l^* + 1, \ldots, l - 12$. Denote the result as $l_p$.

    (c) If there is no result in the previous step, find the earliest $l^*$ in $[l - 23; l - 20]$, such that $\hat{\beta}$ is significantly different from zero at the 99% level for each lead time $l^* + 1, \ldots, l - 20$.

3. (a) If Step 2 has yielded a result for $l_p$, set the length of the adjustment period $p = l - l_p$.

    (b) If Step 2 has not yielded a result, set $p$ equal to the average of the values for the neighboring lead times $l - 1$ and $l + 1$.





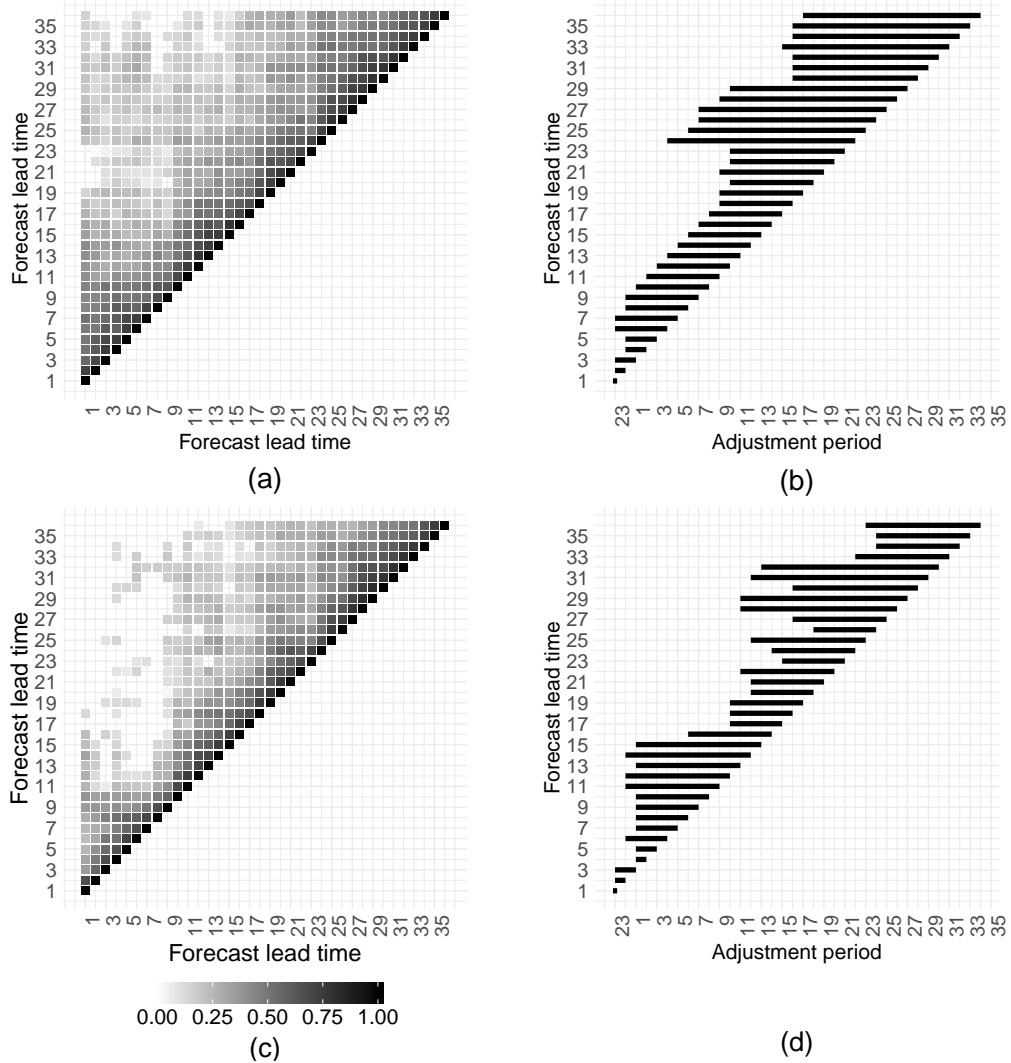

**Figure 2.** (a) Correlation matrix of the EMOS mean error at The Cairnwell. Only correlations significant at the 90% level are shown. (b) Length of the RAFT$_m$ adjustment period for each lead time. (c) As (a), but for the error of an ensemble member drawn from the EMOS predictive distribution. (d) Length of the RAFT$_{ens}$ adjustment period for each lead time.

    (c) If there is still no valid value for $p$, set it to $p = 22$. This corresponds to the longest possible adjustment period.

    The right column of Fig. 2 shows the adjustment periods for the RAFT$_m$ (top) and RAFT$_{ens}$ (bottom) versions at The

Cairnwell. For the ensemble method, the algorithm results in a good approximation of the correlation pattern in Panel (c), but the values of $p$ seem to jump back and forth with increasing lead time. In case of the EMOS mean, the values of $p$ are more consistent across the lead times, but don't necessarily correspond as well to the respective correlation matrix pattern in Panel (a).



Finding the optimal adjustment periods concludes the estimation part of RAFT. The actual adjustment of the predicted forecast error happens in real time once the current model run has finished and the forecasts have been issued. For lead time $l$, the adjustment starts at $l - p + 1$, using the observation recorded at $l - p$, and then continues hourly until $l - 1$. The smaller the gap between $l$ and the time the observation was recorded, the greater the value of the error information and therefore the larger the gain in forecast skill.

In practice, we calculate the observed error according to Eq. (7), plug it into Eq. (11) with the appropriate coefficients $\hat{\alpha}$ and $\hat{\beta}$ and obtain the predictive error $\hat{e}_{t,l}$. Then we can add this forecast to the EMOS mean $m_{t,l}$ for RAFT$_\mathrm{m}$ or the ensemble member $x_{t,l}^{(i)}$, $i = 1, \ldots, 12$ drawn from the EMOS distribution for RAFT$_\mathrm{ens}$:

$$\hat{m}_{t,l} = m_{t,l} + \hat{e}_{t,l} \tag{12}$$
$$\hat{x}_{t,l}^{(i)} = x_{t,l}^{(i)} + \hat{e}_{t,l} \tag{13}$$

Any values of $\hat{m}_{t,l}$ and $\hat{x}_{t,l}^{(i)}$ that become negative during this process are set to zero in order to account for the non-negativity of wind speed. While we can use the RAFT-adjusted mean $\hat{m}_{t,l}$ as a deterministic forecast, the corresponding location parameter $\hat{\mu}_{t,l}$ is needed to evaluate the full distribution. For this purpose, we solve Equations (9) and (10) numerically for $\mu$. This approach can be quite unstable and has to be done carefully so that the resulting distribution is valid. We then combine the new location parameter with the unchanged EMOS variance and thus obtain a predictive distribution. In the case of RAFT$_\mathrm{ens}$, the ensemble members span a discrete distribution. Therefore, we here not only adjust the deterministic forecast, but also simultaneously the spread of the distribution in an adaptive and flow-dependent way.

## 4  Evaluation methods

There is a multitude of evaluation methods available to assess both deterministic and probabilistic forecast skill (see e.g., Thorarinsdottir and Schuhen, 2018). In addition to looking at univariate verification results, we also want to determine the benefit of various combinations of post-processing methods in a multivariate sense.

Proper scoring rules (Gneiting and Raftery, 2007) are useful tools that assign a numerical value to the quality of a forecast and always judge the optimal forecast to have the best score. Usually, they are averaged over a number of forecast cases $n$. In the deterministic case, the root-mean-square error (RMSE) gives an indication about the forecast accuracy of the mean forecast, be it the mean of a distribution or an ensemble mean. It is defined as

$$\mathrm{RMSE}\,(F, y) = \sqrt{\frac{1}{n} \sum_{i=1}^{n} \left(\mathrm{mean}\,(F) - y\right)^2}, \tag{14}$$

where $y$ is the verifying observation corresponding to the predictive distribution $F$.

For evaluating probabilistic forecasts, the CRPS (Matheson and Winkler, 1976) is an obvious choice. Given the score's robustness, it is often used for parameter estimation, as in the two EMOS variants gEMOS and logEMOS described in Sect. 3.1.





A closed form of the CRPS for the truncated Gaussian was derived by Thorarinsdottir and Gneiting (2010) as

$$\text{CRPS}\left(\mathcal{N}^+\left(\mu,\sigma^2\right),y\right) = \sigma \cdot \Phi\left(\frac{\mu}{\sigma}\right)^{-2} \left[\frac{y-\mu}{\sigma}\Phi\left(\frac{\mu}{\sigma}\right)\left\{2\Phi\left(\frac{y-\mu}{\sigma}\right)+\Phi\left(\frac{\mu}{\sigma}\right)-2\right\}\right.$$

(15)

$$\left. + 2\varphi\left(\frac{y-\mu}{\sigma}\right)\Phi\left(\frac{\mu}{\sigma}\right)-\frac{1}{\sqrt{\pi}}\Phi\left(\sqrt{2}\frac{\mu}{\sigma}\right)\right],$$

(16)

where $\Phi$ is the CDF and $\varphi$ the PDF of a standard normal distribution. For the truncated logarithmic distribution, a closed form is also available (Scheuerer and Möller, 2015):

$$\text{CRPS}\left(\mathcal{L}^+\left(\mu,s\right),y\right) = (y-\mu)\left(\frac{2p_y-1-p_0}{1-p_0}\right)$$

(17)

$$+ s\left[\log\left(1-p_0\right)-\frac{1+2\log\left(1-p_y\right)+2p_y\text{logit}\left(p_y\right)}{1-p_0}-\frac{p_0^2\log\left(p_0\right)}{\left(1-p_0\right)^2}\right]$$

(18)

Here, $\text{logit}\left(\cdot\right)$ is the logit function and $p_0 = \Lambda\left(-\mu s^{-1}\right)$ and $p_y = \Lambda\left(\left(y-\mu\right)s^{-1}\right)$ are values of the CDF of the truncated logistic distribution $\Lambda$. To be able to compare all types of forecasts in a fair way, we draw random samples $X_1,\ldots,X_{12}$ from every continuous predictive distribution and evaluate them using the ensemble version of the CRPS:

$$\text{CRPS}_{\text{ens}}\left(X_1,\ldots,X_{12};y\right) = \frac{1}{12}\sum_{i=1}^{12}|X_i-y|-\frac{1}{2\cdot 12^2}\sum_{i=1}^{12}\sum_{j=1}^{12}|X_i-X_j|$$

(19)

Furthermore, we want to assess the level of calibration in a forecast separately, as it is important to prefer the sharpest forecast subject to calibration (Gneiting et al., 2007). To this end, we check the verification rank histogram (Anderson, 1996; Hamill and Colucci, 1997; Talagrand et al., 1997), where we find the ranks of the observation within the forecast ensemble for each forecast case and plot them as a histogram. A flat histogram denotes perfect calibration, while a $\cap$-shaped one points towards overdispersed forecast distributions. Vice versa, a $\cup$ shape means that the forecasts don't exhibit enough spread.

The direct equivalent of the CRPS for multivariate forecasts, the energy score (Gneiting and Raftery, 2007), is defined as

$$\text{ES}\left(F,y\right) = \mathbb{E}_F\left\|X-y\right\|-\frac{1}{2}\mathbb{E}_F\mathbb{E}_F\left\|X-X'\right\|,$$

(20)

where $X$ and $X'$ are independent random vectors drawn from the multivariate distribution $F$, $y$ is the observation vector and $\|.\|$ is the Euclidean norm. If we replace the absolute value in Eq. (19) with the Euclidean norm, we obtain an analogous version of the energy score for ensemble member vectors. It is also possible to evaluate deterministic forecasts in multiple dimensions using the Euclidean error, which we derive from the energy score by replacing the distribution $F$ with a point measure:

$$\text{EE}\left(F,y\right) = \left\|\text{med}_F-y\right\|$$

(21)

The multivariate point forecast $\text{med}_F$ is the spatial median, computed numerically using the R package ICSNP (Nordhausen et al., 2015). It minimizes the sum of the Euclidean distances to the ensemble members.

While the energy score is generally more sensitive to errors in the predictive mean (Pinson and Tastu, 2013), the variogram score proposed by Scheuerer and Hamill (2015) is better at identifying whether the correlation between the components is





correct. In addition to following the authors' recommendation and setting the score's order $p$ to 0.5, we assign equal weights
to all lead times. The variogram score then becomes

$$\text{VS}\,(F,y) = \sum_{i=1}^{36}\sum_{j=1}^{36}\left(\|y_i - y_j\|^p - \mathbb{E}_F\,\|X_i - X_j\|^p\right)^2, \tag{22}$$

where $y_i$ and $y_j$ are the $i$th and $j$th component of the observation vector and and $X_i$ and $X_j$ components of a random vector
distributed according to $F$.

Finally, there are several possibilities to check multivariate calibration, like the multivariate rank histogram (Gneiting et al.,
2008), the band depth histogram and the average rank histogram (both Thorarinsdottir et al., 2016). We choose to use the latter
in this case, as it is less prone to give misleading results than the multivariate rank histogram and more easily interpretable than
the band depth histogram. First, a so-called prerank is calculated, corresponding to the average univariate rank of the vector
components:

$$\rho_S\,(u) = \frac{1}{36}\sum_{i=1}^{36}\text{rank}_S\,(u_i), \tag{23}$$

with $\text{rank}_S\,(u_i)$ being the rank of the $i$th component of a vector $u$ within the combined set of ensemble member and ob-
servation vectors $S = \{x_1,\ldots,x_{12},y\}$. Then the multivariate average rank is the rank of the observation prerank in the set
$\{\rho_S\,(x_1),\ldots,\rho_S\,(x_{12}),\rho_S\,(y)\}$. The interpretation of the average rank histogram mirrors that of the univariate rank histogram
and errors in the correlation structure present themselves in the same way as dispersion errors in the marginal distributions
(Thorarinsdottir and Schuhen, 2018). Visualization of the histograms is taken from Barnes et al. (2019).

## 5  Results

It is the purpose of this paper to investigate if there is a preferred order in applying three different kinds of post-processing
methods. In particular, it will be important to see whether ECC should be run once, like EMOS, subsequent to the end of the
NWP model run, or if it should be continuously applied every time the RAFT adjustment occurs. Therefore, there are two
combinations of methods to be tested: EMOS + RAFT$_m$ + ECC, where RAFT is applied to the EMOS mean only and EMOS +
ECC + RAFT$_{ens}$, where we adjust the EMOS/ECC ensemble members and thus at the same time the prediction of uncertainty.
As we are interested in a comprehensive assessment of the individual combinations' performance, all scores, whether univariate
or multivariate, are of equal importance.

### 5.1  Example forecast

First, we take a look at an example forecast to illustrate how the different RAFT variants work. Fig. 3 shows different forecasts
issued from the 15:00 UTC model run on 30 October 2015 at The Cairnwell, Scotland. The panels on the left hand side depict
the gEMOS + ECC + RAFT$_{ens}$ forecasts, where the mean and prediction interval are obtained by the 12 samples drawn from
the EMOS distribution. On the right hand side, we have gEMOS + RAFT$_m$ forecasts with the mean being the RAFT-adjusted





mean of the EMOS distribution and the variance the unchanged EMOS predictive variance. Here, we show two different stages

in the RAFT adjustment cycle for each combination of post-processing methods. For the top plots, we only apply RAFT once at time $t+1$. This means that all forecasts in the trajectory have been adjusted using the error of the $t+24$ forecast from the model run initialized 24 hours earlier, as long as their corresponding adjustment period allows it. The bottom plots are the results of running the whole RAFT adjustment cycle until the last installment at $t+35$. Consequently, all forecasts have been corrected with the observed error measured two hours earlier and are the most short-term and therefore optimal RAFT forecasts.

In this weather situation, both mean forecasts initially underpredict the wind speed for roughly 12 hours starting from lead time 10, corresponding to the time between 1:00 and 13:00 UTC. A further period of underprediction occurs towards the end of the trajectory, from lead time 28. RAFT is able to recognize these problems quickly and corrects the underprediction quite well, as can be observed in the bottom two panels. However, as the observations are quite jumpy, the sign of the forecast error changes frequently during the adjustment process and the RAFT mean trajectory thus can also exhibit more jumpiness than

the initial EMOS mean. This could be addressed by e.g., adding additional predictors to the RAFT linear regression model in Eq. (11).

There are only minor differences in the mean forecasts between the two post-processing method combinations, while their main difference lies in the derivation of the predictive variance. We can see that the size of the prediction interval for gEMOS + ECC + RAFT$_{\mathrm{ens}}$ changes considerably between the first and the last RAFT adjustment. This is of course because the ensemble,

and therefore the prediction interval spanned by its members, is continuously updated and adjusted in a flow-dependent manner. For example, at the end of the trajectory the ensemble spread in Fig. 3c is much smaller than in Fig. 3d. In the case of gEMOS + RAFT$_{\mathrm{m}}$, the variance is not changed by RAFT and remains at the value originally estimated by EMOS.

## 5.2 Choice of EMOS model

As we tested two versions of EMOS using two different distributions to model the future wind speed observations, we are

interested in which of these, if any, performs better. Initially, we compare the deterministic forecast skill of the EMOS mean and how it is improved by RAFT. In Fig. 4, the RMSE of the gEMOS and logEMOS mean, averaged over all sites and model runs, is shown. Both methods perform very similar, but gEMOS seems to have a slight advantage overall, apart from the first three hours and the last hour. There is a small increase in the RMSE for logEMOS at lead time 23, which is most certainly due to issues finding the minimum CRPS during the EMOS parameter estimation, where all lead times are handled separately.

The ranking of the two EMOS variants is preserved when applying RAFT$_{\mathrm{m}}$ to the EMOS mean forecast. Figure 4a shows the RAFT RMSE if we stopped adjusting the forecasts at $t+15$. This means that all forecasts left of the vertical line have been updated using the observation made two hours earlier, and all forecasts to the right of the line are adjusted using the most recent information available at $t+15$, i.e., the observed error at $t+14$. However, this only applies to those forecasts where lead time 14 lies in the respective adjustment periods. For all other forecasts, the scores for EMOS and RAFT coincide. It is noticeable

that the forecast skill improves significantly as soon as we have information about the error in the current model run at $t+3$. The score remains at about the same level until $t+16$, when it starts to deteriorate, but RAFT still has as the advantage over the EMOS forecasts for another ten hours. In reality, however, we would run RAFT until the end of the forecast cycle, which



**Figure 3.** Example forecast at The Cairnwell initialized at 15:00 UTC on 30 October 2015 for the next 36 hours. The red line corresponds to the RAFT mean forecast, with the shaded area being the $84.6\%$ prediction interval. The verifying observation is indicated by the blue line and the vertical line refers to the current point in time during the RAFT adjustment cycle.

(a) Snapshot of the gEMOS + ECC + RAFT$_{ens}$ forecast taken after RAFT has been applied to the gEMOS + ECC samples once. The prediction interval is spanned by the individually corrected ensemble members. (b) Snapshot of the gEMOS + RAFT$_{m}$ forecast taken after RAFT has been applied to the gEMOS mean once. The prediction interval is based on the gEMOS variance. (c) Same as (a), but RAFT has been applied hourly until the last iteration at $t + 35$. (d) Same as (b), but RAFT has been applied hourly until the last iteration at $t + 35$.


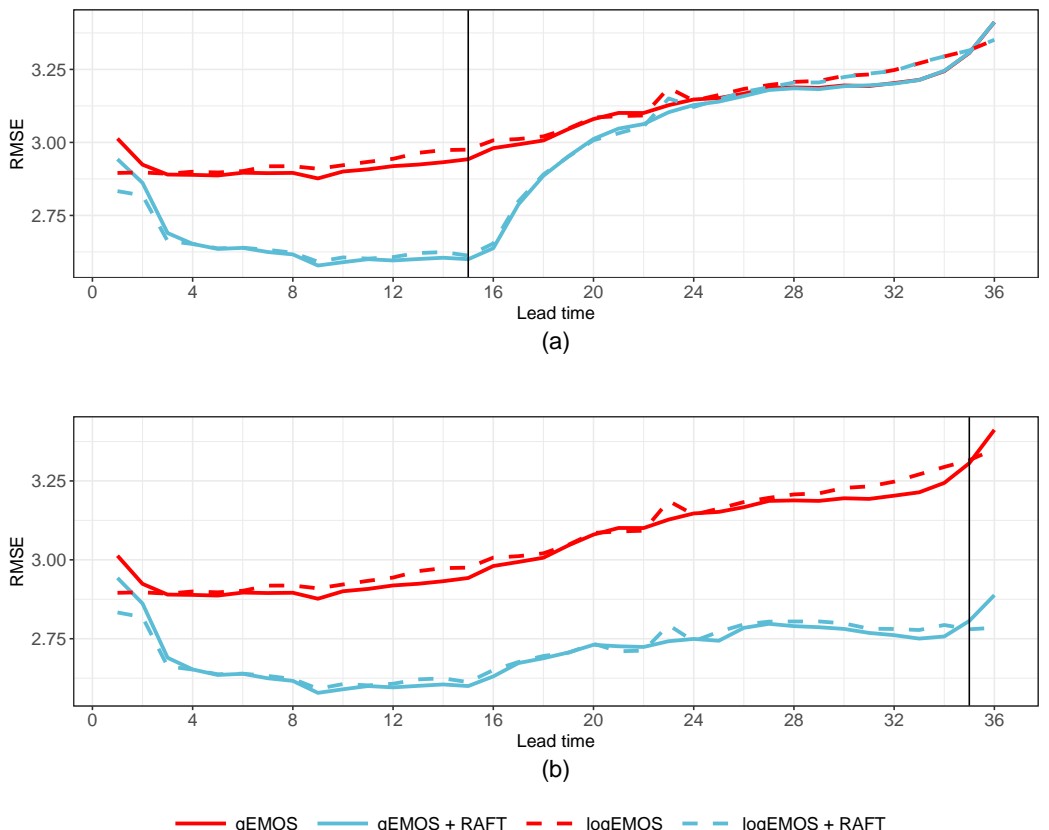

**Figure 4.** RMSE over lead time for gEMOS (red solid line) and logEMOS (red dashed line) mean forecasts, as well as their RAFT$_m$ adjustments (blue solid and blue dashed lines, respectively). The scores are averaged over all model runs and sites in the evaluation set. (a) RAFT is only carried out until the adjustment at $t+15$. (b) RAFT is carried out until its last iteration at $t+35$.

is shown in Fig. 4b. Here, we can see a consistent improvement, especially at large lead times. We also see that the forecasts at lead times 25–26 have more skill than the ones at lead times 1–2, which leads to the conclusion that forecasts from a 24 hour

old model run are for a couple of hours more skillful than forecast from the newest run.

The first column in Table 1 confirms these results. Here, the scores have been aggregated over all lead times, model runs and sites. In this table only scores for RAFT forecasts that have been adjusted one hour previously are shown, i.e., the optimal forecasts. We test the significance of score differences by applying a permutation test based on resampling, as described in Heinrich et al. (2019) and Möller et al. (2013). Both EMOS methods increase the deterministic skill considerably when

compared to the raw ensemble and then are further improved by applying RAFT$_m$. While the RMSE for gEMOS is significantly better than for logEMOS, which we also see in the CRPS, there is almost no difference in the gEMOS + RAFT$_m$ and logEMOS + RAFT$_m$ scores. In terms of the CRPS, logEMOS + RAFT$_m$ has a slight advantage, with the difference being significant at the 95% level.





To confirm that the EMOS forecasts are indeed calibrated, we look at the verification rank histograms in Fig. 5a. While the raw ensemble is very underdispersive, as expected, both gEMOS and logEMOS forecasts are nearly calibrated. One possible reason for the histograms not being perfectly flat is that the here used EMOS models might not be a perfect fit for instantaneous wind speed. Both gEMOS and logEMOS histograms are again very similar, so we compute the coverage of the $84.6\%$ prediction interval created by 12 ensemble members. From the results we see that logEMOS, with a value of $85.18\%$, is much closer to the nominal value than gEMOS with $80.56\%$ and therefore better calibrated. Figure 5b shows the histograms after we apply $\mathrm{RAFT_m}$. Whereas the EMOS variance was on average slightly too small before, is is now a little to big, indicated by the small hump in the middle. This is due to the fact that we don't adjust the variance in this process, but the deterministic skill improves greatly. There is almost no difference in the two histograms, which is also evident in the coverage of the prediction interval, with values of $83.80\%$ and $83.44\%$ for gEMOS and logEMOS, respectively.

In conclusion, there is little difference in the overall performance of the two EMOS variants. While logEMOS has the advantage of being slightly better calibrated, gEMOS shows better scores. After applying RAFT, the two methods are essentially equal. In the following we will therefore only present results from one of the two EMOS versions.

### 5.3 Predictive performance for combinations of post-processing techniques

The main focus of this study is to find out, in which order EMOS, RAFT and ECC should be combined. For RAFT, we employ two different approaches: $\mathrm{RAFT_m}$, where the adjustments are only applied to the EMOS mean and are then combined with the EMOS variance to obtain a full predictive distribution, and $\mathrm{RAFT_{ens}}$, where we adjust individual ensemble members and consequently both mean and spread. In the latter case, ECC is only run once when EMOS has finished, in the first it has be to applied at every RAFT step for the remaining lead times in the forecast run. Therefore the required computing resources depend on the ratio of ensemble members to lead times. In this study, the EMOS + $\mathrm{RAFT_m}$ + ECC combination takes about $33\%$ more time to compute than EMOS + ECC + $\mathrm{RAFT_{ens}}$, however both are with only a few seconds per model run and site computationally very sparse.

When we compare Figures 5b and 5c, is is obvious that the EMOS + ECC + $\mathrm{RAFT_{ens}}$ combination produces forecasts which are slightly less calibrated than EMOS + $\mathrm{RAFT_m}$ forecasts. In fact, the level of calibration deteriorates from the baseline EMOS methods. This also can be deduced from the CRPS values in Table 1, where EMOS + $\mathrm{RAFT_m}$ clearly performs better. The RMSE for both methods is quite similar, so that we can ascribe the discrepancy in the CRPS to the different levels of calibration. Both methods improve the EMOS baseline forecast considerably. In the case of EMOS + $\mathrm{RAFT_m}$, we know this improvement in forecast skill is only due to the adjusted mean forecast, which simultaneously results in better calibrated predictive distributions.

As we are not only interested in the univariate performance, but also in the multidimensional dependencies between the lead times of a forecast trajectory, we look at several multivariate scores (Table 1). The Euclidean error agrees with the univariate RMSE that the best deterministic result can be achieved by applying RAFT last. ECC seems not to have any effect on the scores, which can be expected, as we are only rearranging ensemble members and not necessarily change the multivariate median. The energy score is a measure for the overall skill, but is also more sensitive to errors in the mean forecast. This





**Table 1.** Univariate and multivariate mean scores for different post-processing method combinations, using the final RAFT adjustments one hour before valid time. Bold numbers denote the best score in each column. All score differences are significant at the 95% level, apart from the ones marked with an asterisk, where the pairwise differences between the versions using gEMOS and logEMOS are not significant.

| | RMSE | CRPS | Euclidean Error | Energy Score | Variogram Score |
|---|---|---|---|---|---|
| Raw ensemble | 3.670 | 2.116 | 19.207 | 15.132 | 847 |
| gEMOS | 3.056 | 1.618 | 16.539 | 13.000 | 956 |
| logEMOS | 3.070 | 1.622 | 16.589 | 13.028 | 957 |
| gEMOS + ECC | 3.056 | 1.618 | 16.549 | 12.312 | 812 |
| logEMOS + ECC | 3.070 | 1.622 | 16.607 | 12.356 | 815 |
| gEMOS + RAFT$_m$ | 2.713* | 1.445 | 15.045 | 11.943 | 899 |
| logEMOS + RAFT$_m$ | 2.714* | **1.443** | 15.029 | 11.913 | 897 |
| gEMOS + RAFT$_m$ + ECC | 2.713* | 1.445 | 15.049 | 11.175 | **784*** |
| logEMOS + RAFT$_m$ + ECC | 2.714* | **1.443** | 15.033 | 11.165 | **784*** |
| gEMOS + ECC + RAFT$_{ens}$ | **2.708*** | 1.483 | 15.024* | **11.164*** | 786 |
| logEMOS + ECC + RAFT$_{ens}$ | 2.709* | 1.482 | **15.023*** | 11.166* | 787 |

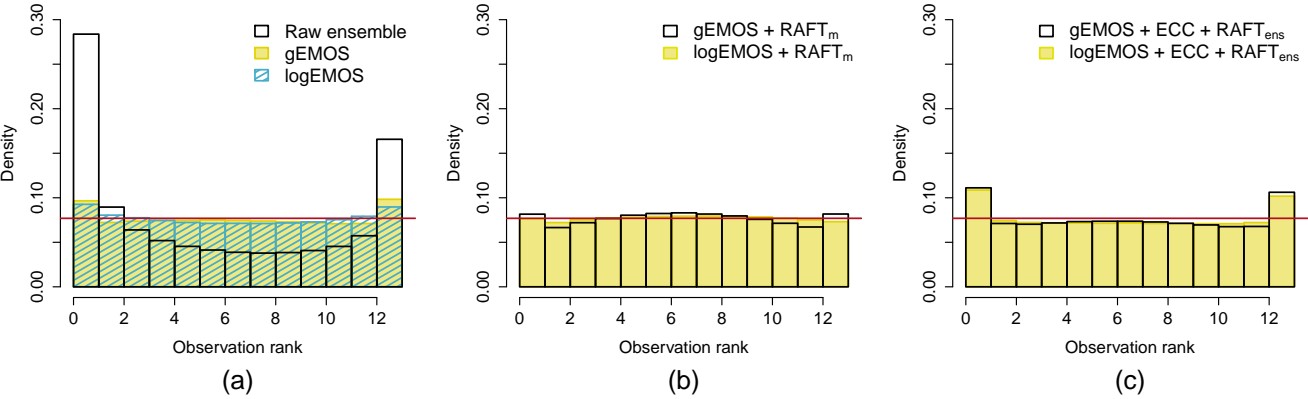

**Figure 5.** Verification rank histograms for different forecasting methods, aggregated over all sites, model runs and lead times. RAFT histograms are based on the final adjustment for each lead time.

is the reason why RAFT$_m$ manages to improve the energy score as compared to EMOS + ECC, while the variogram score deteriorates. Note that both scores are reduced when we reintroduce the temporal correlation structure by applying ECC to the

EMOS + RAFT$_m$ forecasts. Although the energy and variogram scores for EMOS + ECC + RAFT$_{ens}$ and EMOS + RAFT$_m$




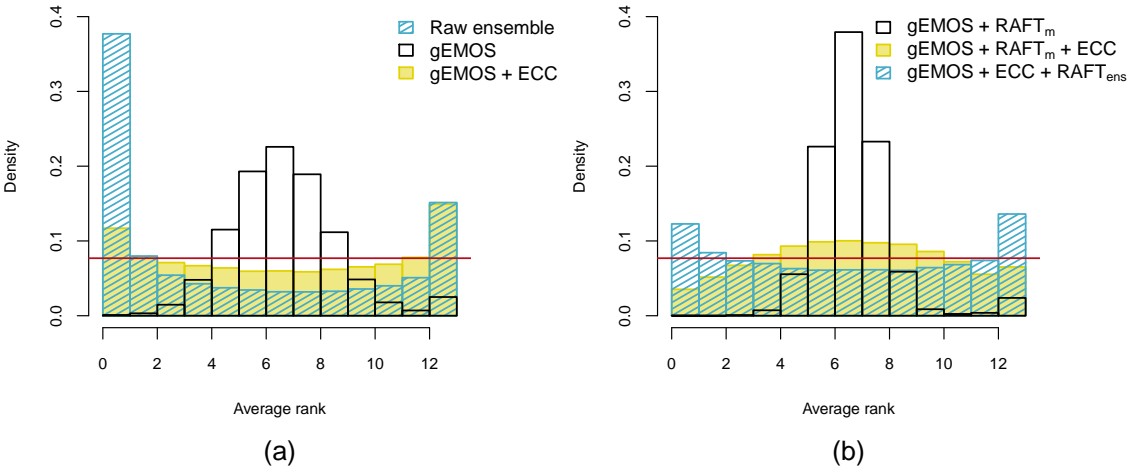

**Figure 6.** Average rank histogram for different combinations of post-processing methods. Data points are aggregated over all sites, model runs and lead times. All RAFT forecasts have been adjusted using the observation measured two hours earlier.

+ ECC are very close, the two scores prefer different post-processing method combinations. While the energy score judges the method to be the best where we apply ECC first, which also has the best RMSE and Euclidean error, the variogram score assigns the lowest value to the better calibrated EMOS + RAFT$_m$ + ECC. From this we can conclude that RAFT$_{ens}$ manages to preserve the multivariate correlation structure throughout its multiple iterations.

The average rank histograms in Fig. 6 confirm that without applying ECC, the EMOS and EMOS + RAFT$_m$ forecasts are very uncalibrated. Both the EMOS + ECC and EMOS + ECC + RAFT$_{ens}$ combinations are underdispersed, as was also seen in the univariate histograms. On the other hand, the EMOS + RAFT$_m$ + ECC forecast ranks form a hump-like histogram. This is due to the correlation between the components being too weak, as the band depth histogram confirms (not shown; see Thorarinsdottir et al., 2016).

In order to investigate further the optimal order of operation when applying multiple post-processing methods, we look at how the scores develop with every step in the RAFT process. While the scores in Table 1 are computed using the final RAFT installment at $t + 35$, where all forecasts have been adjusted using the observation made two hours earlier, Fig. 7 shows the CRPS, energy score and variogram score computed at each RAFT iteration for the gEMOS + ECC + RAFT$_{ens}$ and gEMOS + RAFT$_m$ + ECC forecasts. From Fig. 7a, it is clear that EMOS + RAFT$_m$ + ECC performs best in terms of the CRPS, with

the gap between the two combinations widening with increasing number of RAFT adjustments. As we have also seen that the RAFT$_m$ version is better calibrated than the RAFT$_{ens}$ one, this means that the CRPS here puts more weight on the calibration being correct than on the slightly better deterministic forecast (see the RMSE in Table 1) in the RAFT$_{ens}$ case. This is surprising, given that the CRPS is usually more sensitive to the error in the forecast mean.


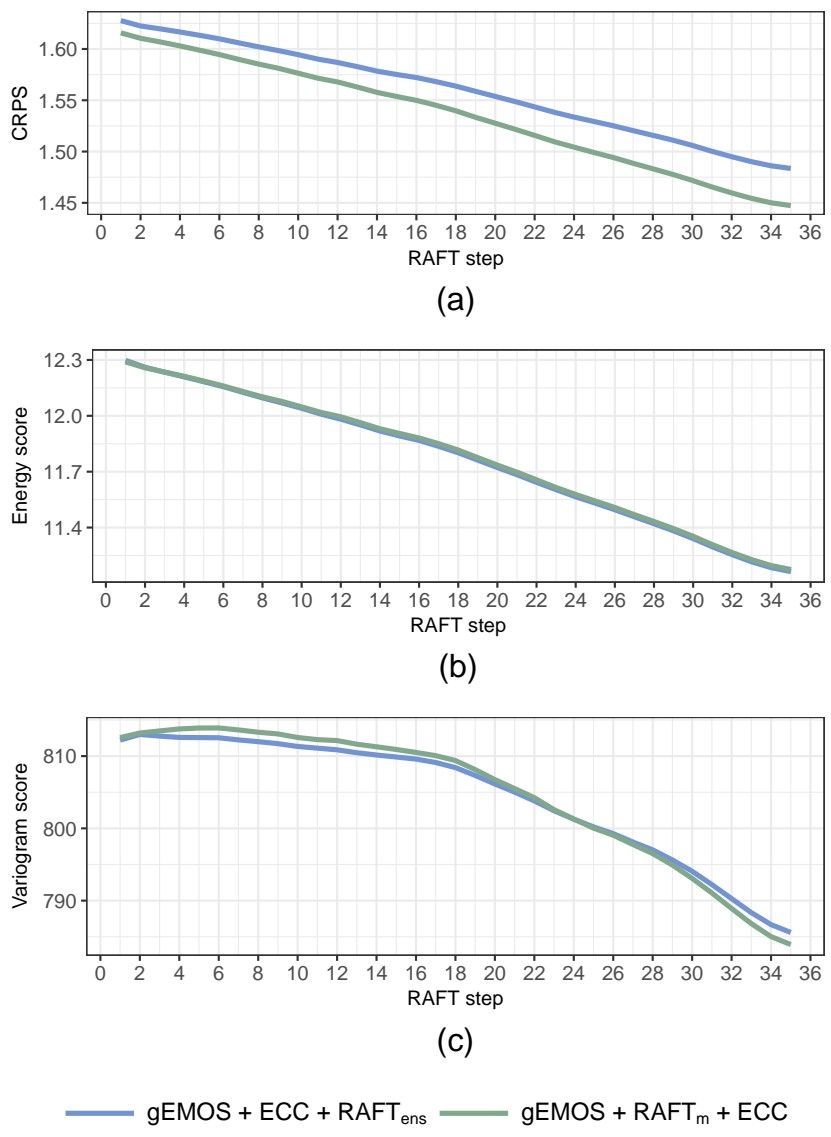

gEMOS + ECC + RAFT$_{ens}$     gEMOS + RAFT$_m$ + ECC

**Figure 7.** Mean CRPS, energy score and variogram score for every step in the RAFT$_m$ and RAFT$_{ens}$ process. Scores are averaged over all lead times, sites and model runs.

While the CRPS results show a clear pattern, it is not as straightforward for its multivariate analogue, the energy score. The mean score decreases with every RAFT adjustment, as expected, but there is no discernible difference in the performance of the two post-processing method combinations. The most complex picture emerges in the case of the variogram score, where the ranking of the two combinations actually switches around RAFT iteration 24. The variogram score is better at detecting incorrect correlation structures than the energy score, so one possible explanation would be that EMOS + ECC + RAFT$_{ens}$ is

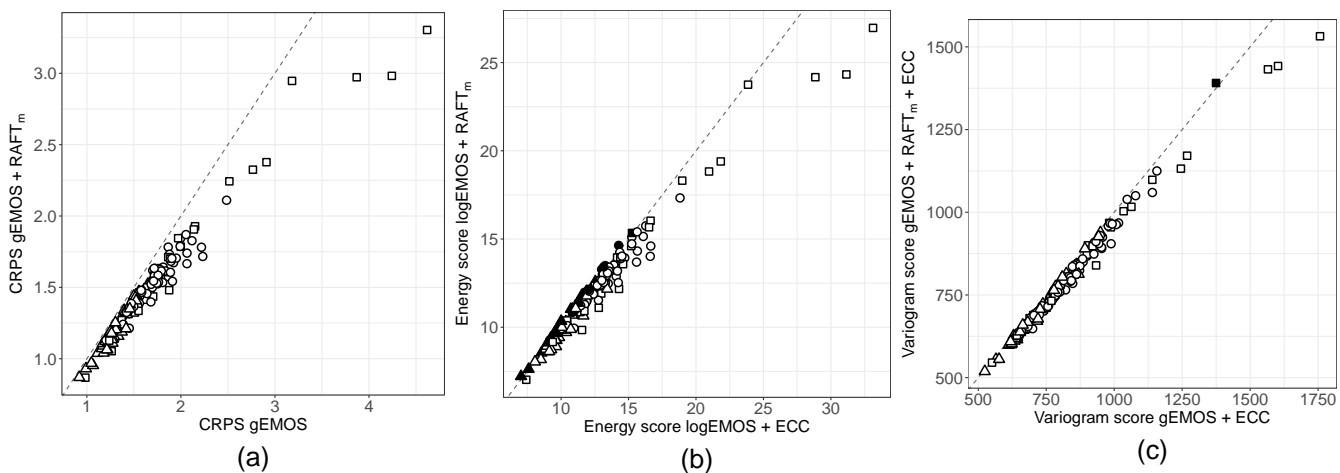

Site type: ○ coastal △ inland □ mountain

**Figure 8.** (a) CRPS of gEMOS + RAFT$_m$ forecasts against the CRPS of gEMOS forecasts for individual sites. (b) Energy scores of logEMOS + RAFT$_m$ forecasts against energy scores of logEMOS + ECC forecasts for individual sites. (c) Variogram scores of gEMOS + RAFT$_m$ + ECC forecasts against variogram scores of gEMOS + ECC forecasts for individual sites. The scores are averaged over all lead times and model runs. RAFT forecasts are taken from the final iteration. Filled symbols denote that the score on the y-axis is lower than the one on the x-axis, empty symbols denote the opposite.

initially good at retaining the appropriate correlations, but that ability weakens over time. Conversely, ECC is applied after
every iteration of RAFT$_m$, which might explain the better variogram scores towards the end of the process. However, we have
observed in Fig. 6 that the correlation structure at the last iteration is still too weak. It should also be noted that the variogram
score for EMOS + RAFT$_m$ + ECC deteriorates slightly at the beginning.

Finally, we want to investigate the homogeneity of the scores across the different locations and highlight some interesting
results for particular sites. In Fig. 8a, we see that RAFT$_m$ improves the CRPS for all sites as compared to the EMOS baseline.
That means that the method where the variance is not adjusted increases the deterministic and probabilistic forecast skill at
all sites. As we have seen from the univariate histograms in Fig. 5, even the calibration is improved. Figure 8b shows how a
reduction in the mean error can have a large effect on the energy score. At 37 sites, the energy score for logEMOS + RAFT$_m$ is
actually lower than the one for logEMOS + ECC. The former forecasts are lacking any form of temporal coherency among lead
times, so here the deterministic improvement exceeds any benefit from reintroducing the ensemble's correlation information.
Judging from Fig. 8c, a case can be made for a site-specific RAFT approach. The mean variogram score for the gEMOS +
RAFT$_m$ + ECC forecasts at The Cairnwell, Scotland is higher than the score for gEMOS + ECC, meaning that we are not able
to make any improvements by applying RAFT$_m$ and that there are local effects not resolved by the RAFT model.





## 6    Conclusions

Our goal was to find out in which order post-processing methods pertaining to different stages in the forecasting process
should be applied. We look at three techniques, each with a different objective. EMOS is a versatile method aiming to calibrate
ensemble output as soon as the model run is finished, based on the ensemble's performance over the last 40 days. There are two
candidates for wind speed calibration: gEMOS uses a model based on truncated Gaussian distributions and logEMOS a model
based on truncated logarithmic distributions. It turns out that both models produce very similar results, with gEMOS having
slightly better scores and logEMOS being a little closer to perfect calibration. Therefore it is advised to test both methods for
the data set at hand and to check which distribution gives a better fit.

The second technique, ECC, restores the multivariate dependency structure present in the ensemble forecasts to the EMOS
predictions. While conceptually and computationally easy to implement, the success of ECC relies on the NWP model getting
the physical, spatial and temporal correlations between the components right. Making use of the part of a forecast trajectory
that has already verified, RAFT is based on the concept that an observed error will provide information about the expected error
at not-yet-realized lead times. It can be applied either to the forecast mean only ($RAFT_m$) or to a set of ensemble members
($RAFT_{ens}$) in order to adjust both predictive mean and variance.

In essence, there are two feasible options when combining these three methods: EMOS + ECC + $RAFT_{ens}$ and EMOS +
$RAFT_m$ + ECC. Overall, their performance might be very similar, but there are subtle differences which can lead to preferring
one method over the other. The EMOS + $RAFT_m$ + ECC variant produces a lower CRPS and has better univariate calibration,
although this is most likely a feature of this forecasting system only, where the EMOS forecasts are underdispersive. Naturally,
the $RAFT_{ens}$ adjusted predictive variance becomes smaller with every RAFT step, as predictability usually increases with
a shrinking forecast horizon. This, however, leads to the respective distributions still being underdispersed and not able to
counterbalance the deficit of the EMOS forecasts.

If multivariate coherency is of particular importance, e.g., to create plausible forecast scenarios, the EMOS + ECC + $RAFT_{ens}$
turns out to be the better choice, as is beats its alternative in terms of the energy score, the Euclidean error and also the RMSE,
while there is only very little difference in the variogram score. It is also more versatile and should be preferred for NWP
ensembles exhibiting very different calibration characteristics than MOGREPS-UK.

Therefore, it is necessary to study every forecasting scenario closely, monitor how calibration methods like EMOS affect
the forecast skill and identify potentially remaining deficiencies. As a rule of thumb, it can be said that the post-processing
method designed to address one's particular area of interest, whether univariate or multivariate, should be applied first. So far,
we do not adapt RAFT to optimize forecasts at individual sites. A model tailored to specific local characteristics could involve
changing the algorithm for finding the adjustment period or adding suitable predictors to the linear model. Also, particular
attention should be paid if the focus lies on a specific subset of lead times, or if the forecasts have to be irrevocably issued at a
certain point in time, as the ranking of methods can change during the RAFT process.



*Author contributions.*  NS designed and executed the study, as well as composed this manuscript.

*Competing interests.*  The author declares that there are no conflicts of interest.

*Acknowledgements.*  The author would like to thank the IMPROVER team at the UK Met Office for the opportunity to work with MOGREPS-
UK data and Thordis L. Thorarinsdottir for numerous helpful discussions and feedback. This work was supported by the Research Council
of Norway through grant nr. 259864 "Stipendiatstillinger til Norsk Regnesentral".



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
