# Peer review of "Order of operation for multi-stage post-processing of ensemble wind forecast trajectories"

_Nonlinear Processes in Geophysics, 2019_

## Referee Comment (RC1) · Anonymous Referee #1 · 24 Nov 2019

This manuscript proposes and evaluates an extension of the RAFT (rapid adjustment of forecast trajectories) approach, which updates previously generated probabilistic forecasts based on new observations that become available as time elapses. While the original approach applied the RAFT approach only to the predictive mean, the extension proposed here combines the RAFT approach with ECC (ensemble copula coupling) to test whether it is beneficial to apply RAFT to individual, post-processed ensemble members. A comprehensive comparison with the alternative strategy (applying RAFT only to the predictive mean and generate calibrated ECC ensembles as the final step of the calibration procedure) is performed using MOGREPS-UK ensemble wind speed forecasts and wind speed observations at 152 SYNOP observation sites across the British Isles.

The manuscript is well written and the RAFT extension proposed here is interesting. Conclusions are well supported by the verification results presented in various figures and tables. Some minor clarifications are required as discussed below.

Specific comments:

161: For RAFT_ens, were separate correlation matrices calculated for each member or were data pooled across all members and a single correlation matrix estimated and used for all members?

191: The explanation here is somewhat confusing. What exactly is averaged here? Please add some more detailed explanation. Also, while I don't fully understand in which sequence the different lead times are being processed, I am surprised that both the preceding and subsequent lead times are assumed to have been processed at this point. Please add some explanation for this as well.

242: I am not happy with the formulation 'A flat histogram denotes perfect calibration'. A flat histogram is a necessary but not sufficient for calibration, and examples can be constructed where uncalibrated forecasts yield near flat histograms (e.g. Hamill, 2001, or Thorarinsdottir et al., 2016, their Fig. 7).

368 'From this we can conclude ...': I don't disagree with the conclusion in general but I find this pair of sentences logically confusing. If EMOS + RAFT_m + ECC results in a better score, how does this imply that RAFT_ens preserves the multivariate correlation structure? Please clarify.

371: I would be careful with the use of the term 'underdispersed' in connection with the average rank histogram. What we see is a U-shape, but the discussion in Thorarinsdottir et al., 2016 suggests that this can be caused by miscalibration other than underdispersion of the ensemble. I'd refer to the univariate histograms first as their interpretation is less ambiguous.

383: Can the author provide a reference for the claim that 'the CRPS is usually more

sensitive to the error in the forecast mean'? This is not obvious to me since the CRPS - unlike RMSE or Euclidean error - assesses the calibration of the full distribution.

Fig. 8b: While I agree that at this point it is not required to always show results for both gEMOS and logEMOS given that it has been found previously that their performance is comparable, I wonder why the author chose to alternate between the two. This could create the impression that figures where hand picked to support a particular conclusion. I'd either consistently use gEMOS or logEMOS, or provide the underlying rationale if there is indeed a good reason to consider gEMOS in one and logEMOS in another context.

Language and typos:

346-347: it has be to applied -> it has to be applied

351: is is -> it is

References:

Hamill, T. M., 2001: Interpretation of rank histograms for verifying ensemble forecasts. Mon. Wea. Rev., 129, 550-560.

Thorarinsdottir, T. L., Scheuerer, M., and Heinz, C., 2016: Assessing the calibration of high-dimensional ensemble forecasts using rank histograms. J. Comput. Graph. Stat., 25, 105–122.

---

## Referee Comment (RC2) · Anonymous Referee #2 · 26 Nov 2019

The author discusses a generalization of the recently proposes rapid adjustment of forecast trajectories (RAFT) approach of improving the previously obtained predictive distributions using the most recent forecast error information. While in the original work [1] just the mean of the ensemble model output statistics (EMOS) predictive distribution for temperature is adjusted by RAFT, here a more complex question is investigated. After calibration of wind speed ensemble forecasts with two different EMOS models, RAFT is extended with ensemble copula coupling (ECC) in order to account for the dependencies between forecasts at different lead times. The main question of the paper is to decide whether after an EMOS calibration it is more beneficial to apply first RAFT to adjust the EMOS mean and then ECC, or consider ECC first and apply RAFT to the obtained simulated forecasts. The presented results are definitely new

and interesting and in general, the paper is well written and easy to read. However, there are some inaccuracies requiring clarification:

1. P3,L82: The name "non-homogeneous Gaussian regression" is misleading, as there are several EMOS approaches were the predictive distribution is non-Gaussian. Use simply "non-homogeneous regression".

2. P5,L100: $\sigma^2$ is not the variance of the truncated normal distribution. It is a scale parameter.

3. P5,L116: Is there any explanation why the optimization procedure is more stable for wind speed forecasts given in knots?

4. P15,L331: I don't think that the slight deviation of the PIT histograms from uniformity is coming from the use of instantaneous wind speed data. E.g. in [2], where in the case studies both maximal and instantaneous wind speed is considered, the truncated normal EMOS model results in rather similar PIT histograms.

5. The blue and green lines of Figure 7 are hard to distinguish in BW, a different choice of colors or using solid and dashed lines would be better.

References:

1. Schuhen, N., Thorarinsdottir, T. L. and Lenkoski, A. (2019) Rapid adjustment and post-processing of temperature forecast trajectories. arXiv:1910.05101.

2. Baran, S., Lerch, S. (2015) Log-normal distribution based EMOS models for probabilistic wind speed forecasting. Q. J. R. Meteorol. Soc. 141, 2289-2299.

---

## Author Comment (AC1) · 10 Dec 2019

**Response to R1**

*We thank the reviewer for their thoughtful comments and overall positive assessment of our paper. Please find below a point-by-point response to the comments.*

Specific comments

1. 161: For RAFT$_{ens}$, were separate correlation matrices calculated for each member or were data pooled across all members and a single correlation matrix estimated and used for all members?

   *The correlation matrices for RAFT$_{ens}$ are unique to the ensemble members. We added a sentence to the revised manuscript to clarify this.*

2. 191: The explanation here is somewhat confusing. What exactly is averaged here? Please add some more detailed explanation. Also, while I don't fully understand in which sequence the different lead times are being processed, I am surprised that both the preceding and subsequent lead times are assumed to have been processed at this point. Please add some explanation for this as well.

   *We added some text to the description of the algorithm that makes it clear that Step 3 is only carried out after Step 2 has finished for all lead times and that in 3(b)*

*we are averaging the values for the adjustment period length of the neighboring lead times.*

3. 242: I am not happy with the formulation 'A flat histogram denotes perfect calibration'. A flat histogram is a necessary but not sufficient for calibration, and examples can be constructed where uncalibrated forecasts yield near flat histograms (e.g. Hamill, 2001, or Thorarinsdottir et al., 2016, their Fig. 7).

   *This is of course correct. We changed the text to reflect that a flat histogram is not a sufficient condition for calibration.*

4. 368: 'From this we can conclude ...': I don't disagree with the conclusion in general but I find this pair of sentences logically confusing. If EMOS + RAFT$_m$ + ECC results in a better score, how does this imply that RAFT$_{ens}$ preserves the multivariate correlation structure? Please clarify.

   *We changed this sentence to "The almost identical variogram scores suggest that RAFT$_{ens}$ manages to preserve the multivariate correlation structure throughout its multiple iterations." The variogram score is sensitive to misspecifications in the multivariate correlation structure. Thus the results indicate that applying RAFT$_{ens}$ does not destroy the dependencies between lead times, as it performs similar to RAFT$_m$ + ECC, which follows the ensemble's multivariate structure.*

5. 371: I would be careful with the use of the term 'underdispersed' in connection with the average rank histogram. What we see is a U-shape, but the discussion in Thorarinsdottir et al., 2016 suggests that this can be caused by miscalibration other than underdispersion of the ensemble. I'd refer to the univariate histograms first as their interpretation is less ambiguous.

   *We changed the text here, so that it is clear that we consulted the band depth histogram to eliminate other causes of miscalibration than underdispersion. The*

*same holds for our interpretation of the EMOS + RAFT$_m$ + ECC average rank histogram as the correlation between components being too weak.*

6. 383: Can the author provide a reference for the claim that 'the CRPS is usually more sensitive to the error in the forecast mean'? This is not obvious to me since the CRPS- unlike RMSE or Euclidean error - assesses the calibration of the full distribution.

*We added text to the revised manuscript that this follows from Figure 4 in Friederichs and Thorarinsdottir (2012) and is also the case for the multivariate version of the CRPS, the energy score (Pinson and Tastu, 2013).*

7. Fig. 8b: While I agree that at this point it is not required to always show results for both gEMOS and logEMOS given that it has been found previously that their performance is comparable, I wonder why the author chose to alternate between the two. This could create the impression that figures where hand picked to support a particular conclusion. I'd either consistently use gEMOS or logEMOS, or provide the underlying rationale if there is indeed a good reason to consider gEMOS in one and logEMOS in another context.

*The reviewer is correct in that using the logEMOS plot here might create an impression we did not intend to convey. We have switched Figure 8b for the gEMOS version, which looks almost identical.*

8. 346-347: it has be to applied ->it has to be applied

*This has been corrected in the revised manuscript.*

9. 351: is is ->it is

*This has been corrected in the revised manuscript.*

**References**

Friederichs, P. and Thorarinsdottir, T. L.: Forecast verification for extreme value distributions with an application to probabilistic peak wind prediction, *Environmetrics*, 23, 579–594, https://doi.org/10.1002/env.2176, 2012.

Pinson, P. and Tastu, J.: Discrimination ability of the energy score, Technical report, Technical University of Denmark (DTU), 2013.

---

## Author Comment (AC2) · 10 Dec 2019

**Response to R2**

*We thank the reviewer for their thoughtful comments and overall positive assessment of our paper. Please find below a point-by-point response to the comments.*

Specific comments

1. P3,L82: The name "non-homogeneous Gaussian regression" is misleading, as there are several EMOS approaches were the predictive distribution is non-Gaussian. Use simply "non-homogeneous regression".

   *This has been corrected in the revised manuscript.*

2. P5,L100: $\sigma^2$ is not the variance of the truncated normal distribution. It is a scale parameter.

   *This is correct and has been changed at multiple instances in the document.*

3. P5,L116: Is there any explanation why the optimization procedure is more stable for wind speed forecasts given in knots?

   *While we have not investigated this in detail, we think it occurs because the CRPS optimization routine is less stable if the wind values are smaller and thus closer*

*to zero, the cut-off point for the distribution. Our results showed very inconsistent EMOS parameters for some lead times, as compared to its neighbours. This resulted in some unrealistic jumps in forecast skill between lead times.*

4. P15,L331: I don't think that the slight deviation of the PIT histograms from uniformity is coming from the use of instantaneous wind speed data. E.g. in [2], where in the case studies both maximal and instantaneous wind speed is considered, the truncated normal EMOS model results in rather similar PIT histograms.

   *We thank the reviewer for making us aware of this and will omit this sentence in the revised manuscript.*

5. The blue and green lines of Figure 7 are hard to distinguish in BW, a different choice of colors or using solid and dashed lines would be better.

   *We changed one of the lines in Figure 7 to dashed, so that it should now be possible to differentiate between the two lines in both color and BW.*

---

## Author Response (AR2)

**Author's response to the editor**

*We thank the editor for his comments and for accepting our paper for publication.*

1. p 6 /l142: already been verified

   *We changed this sentence slightly to "... using information from the part of the forecast trajectory that has already realized."*

2. p7/l185: I assume $l^* + 24$ of the previous model run?

   *In this estimation step, we only need to make sure that lead time 23 is followed by lead time $0, 1, 2, \ldots$. When applying the RAFT model to adjust the forecasts, this then means using the corresponding forecast from the previous model run. We have added a sentence to clarify this.*

3. p7/l191: Shouldn't this result also be denoted as $l_p$?

   *This is correct and was added to the text.*

[revised manuscript text omitted]